# OpenReview forum: "VeRO: A Harness for Agents to Optimize Agents"
_ICML.cc/2026/Conference — ICML 2026 regular_

### Official Review · Reviewer_Z4n6 · 2026-03-09

**Soundness:** 3
**Presentation:** 3
**Significance:** 3
**Originality:** 3
**Overall Recommendation:** 4
**Confidence:** 4

**Summary:**

VeRO introduces a framework harness and benchmark for agent optimization. While for most tasks, the harness is static, VeRO proposes that we can optimize the harness as well and establishes a framework for doing so (e.g. the prompts, tools, parameters, etc.). Their contributions are the following: 1) a reproducible evaluation harnesses, which uses Git worktrees for snapshots and enforces compute budgets. 2) a benchmark suite to support the work, 3) a detailed empirical analysis to show e.g. that optimization is sensitive to instructions.

**Compliance With Llm Reviewing Policy:**

Affirmed.

**Key Questions For Authors:**

No questions for the authors.

**Limitations:**

Yes

**Strengths And Weaknesses:**

Strengths:
1) One major strength of VeRO is the novelty. While agent frameworks have been well established and critical, little work has been done on optimizing the frameworks themselves. VeRO is an novel framework that tackles this problem, which hasn't been done in research before and can be a valuable new research area for the community.
2) VeRO also allows the agent to show what optimizations have been doing which gives insight into how the agent is being modified.
3) There is a comprehensive empirical study across a variety of benchmarks showing performance increases above the board. Along with this, the authors give analysis on the strategy behind the optimizers changes.
Weaknesses:
1) It's noted that improvements on some types of tasks (e.g. tool-use) can cause regression in other domains. It would be helpful if there were more insights here, as this is expected but not ideal. For example, can we optimize multiple benchmarks at the same time?
2) A human baseline, even small scale (e.g. done by the authors) could be insightful. For example, an annotator can take a look at the prompt and failure cases and try to fix mistakes the model is making, which is a common workflow for improving (static) agent harnesses.

---

> ### Author Rebuttal · Authors · 2026-03-30
>
> We thank the reviewer for recognizing VeRO's novelty and the value of the interpretability analysis.
>
> **W1/Q1:** Our case study provides concrete evidence. Under *Cookbook+Reasoning*, Pawn iteration 3 achieved +5.75% on GAIA but regressed -17.78% on SimpleQA. Git diff inspection reveals: the optimizer added a complex multi-step verification tool that improved multi-hop reasoning (GAIA) but introduced unnecessary overhead for simple factual queries (SimpleQA).
>
> | Run | FACTS Δ | GAIA Δ | SimpleQA Δ |
> |---|---|---|---|
> | Cookbook+Reasoning Pawn iter3 | -5.39% | +5.75% | **-17.78%** |
> | Minimal Pawn iter1 | -5.96% | -2.30% | **-17.78%** |
>
> **Pattern:** Task-specific optimizations (tool additions, complex verification) improve performance on tasks requiring those capabilities but add overhead that degrades simpler tasks. Our recommendation: multi-task validation during optimization. When the optimizer evaluates on a single training set, it lacks signal about cross-task effects (Section 5.3).
>
> **Can we optimize on multiple benchmarks simultaneously?** Yes — if multiple tasks are aggregated into a single dataset, VeRO supports this directly (the case study evaluates on three datasets simultaneously). What VeRO does not currently support is multi-objective optimization with a vectorial objective (e.g., separate scores per task with Pareto trade-offs). This is an important direction for future work.
>
> **W2/Q2:** We acknowledge this limitation (Appendix A.4). A meaningful human baseline requires matched constraints: same budget (B=8), same tools, no test data access, same splits. Public leaderboards reflect months of unconstrained engineering — not comparable to B=8 sessions. We plan to conduct a developer study where expert developers optimize target agents under VeRO's protocol, providing a human upper bound for future work.

---

> > ### Author Rebuttal · Reviewer_Z4n6 · 2026-04-02
> >
> > Thanks to the authors for the rebuttal. I maintain my positive score.

---

> > > ### Author Response · Authors · 2026-04-07
> > >
> > > We thank Reviewer Z4n6 for confirming that our responses addressed their concerns.

---

### Official Review · Reviewer_EH22 · 2026-03-10

**Soundness:** 2
**Presentation:** 3
**Significance:** 3
**Originality:** 3
**Overall Recommendation:** 4
**Confidence:** 4

**Summary:**

This paper introduces VeRO, an evaluation harness for studying agents that optimize other agents through edit–execute–evaluate loops. The paper formalizes target-agent optimization as a budgeted search problem over agent code, then proposes a concrete infrastructure with versioned snapshots, gated evaluation, structured traces, and permission control.

**Compliance With Llm Reviewing Policy:**

Affirmed.

**Key Questions For Authors:**

To be honest, the main text of this paper is really hard to support "support research on agent optimization as a core capability for coding agents" as described in abstract & intro, why authors write such kind of sentences without including much SWE or coding tasks.

**Limitations:**

- Missing recognized references like GPTSwarm [1] and DyLan [2] which is recognized as important pioneer optimization solutions before TextGrad, ADAS, etc.

[1] Gptswarm: Language agents as optimizable graphs

[2] A Dynamic LLM-Powered Agent Network for Task-Oriented Agent Collaboration

**Strengths And Weaknesses:**

### Strengths

- Optimizing agents-as-code is becoming increasingly relevant to the practical applications, especially coding agents have become more are more popular.

- Evaluation protocol is important, and the paper proposed an interesting point to solve this issue.

### Weaknesses

- The primary benchmark averages over only $N = 3$ iterations/configuration, and many conclusions about optimization dynamics are based on small numbers of runs. Could the authors explain wether these setup are enough to demonstrate the main claim?

- The practical objective is framed as maximizing lift over a baseline on training data, while model selection is based on validation performance, and test evaluation is limited in some analyses to the initial and best-validation commits.

-  Two optimizers consume very different numbers of tokens, shell calls, or external API credits between evaluation calls, then counting only evaluator invocations may not be enough for a fair comparison.

---

> ### Author Rebuttal · Authors · 2026-03-30
>
> We thank the reviewer for recognizing the practical relevance of VeRO and its evaluation protocol.
>
> **W1/Q1:** We agree that more iterations would tighten confidence intervals. The primary constraint is cost: each optimization run involves (B) full-dataset evaluations plus optimizer inference. Our 137 runs represent substantial compute. The consistent trends across tasks (tool-use gains, reasoning stagnation, best commits found early) provide confidence in the experimental infrastructure and directional findings.
>
> **W2:** This follows standard ML protocol: **optimize** on training data (optimizer sees training instances and scores), **select** best commit via validation performance, **report** on held-out test where available. For GPQA, which lacks a validation split, we select on training performance — an acknowledged overfitting risk (see also our response to A9Sp W4/Q2).
>
> **W3:** Acknowledged in Appendix A.4. Budget (B) counts evaluation invocations, not total compute. Table 8 reports runtime per sample as a partial proxy. Between configurations, compute differences may be substantial — e.g., *Orchestrator* can re-write entire code files (more tokens) while *Resources Only* can only make targeted edits (fewer tokens).
>
> **Q4:** There are two distinct roles in VeRO: the *optimizer* (a coding agent) and the *target agent* (which solves a downstream task). The optimization task *is* a coding task — the optimizer reads Python, diagnoses failures from execution traces, writes tools, edits prompts, and manages code through Git. The target agent's task (GAIA, GPQA, etc.) need not be a coding task. Including coding *target tasks* (e.g., SWE-Bench) would increase task diversity but is orthogonal to the paper's core contribution: evaluating coding agents' ability to optimize other agents through code.
>
> **Why no SWE-Bench or TerminalBench?** Both were excluded for practical reasons: SWE-Bench instance evaluation takes 5–15 minutes (Docker setup + test execution); TerminalBench similarly requires container-based execution. At B=8, a single SWE-Bench optimization run would take several hours, making systematic comparison across 7 configurations challenging. VeRO is task-agnostic and can accommodate both with larger compute budgets.
>
> **Reasoning tasks are included by design.** Our benchmark intentionally spans the tool-use vs. reasoning axis. Tool-use tasks (GAIA, Retail, SimpleQA) show +13–19% gains; reasoning tasks (GPQA, MATH) show +1–5%. This establishes that *current agent optimization primarily improves scaffolding, not reasoning* — a finding that would be invisible without the reasoning tasks and is useful for practitioners deciding where to invest optimization effort.
>
> **Revision:** We will clarify the optimizer-vs-target distinction in the introduction and revise the abstract phrasing.
>
> **Q5:** Thank you. **Revision:** We will add GPTSwarm (Zhuge et al.) and DyLan (Liu et al.), both relevant as optimization approaches treating agent topologies as the search space, complementary to VeRO's agent-as-code focus.

---

> > ### Author Rebuttal · Reviewer_EH22 · 2026-04-02
> >
> > My concern has partially solved.  I will maintain my original score.

---

> > > ### Author Response · Authors · 2026-04-07
> > >
> > > Thank you for your engagement during the discussion period.
> > >
> > > Since our initial rebuttal, we have conducted an additional experiment relevant to your Q4 (regarding coding tasks): we ran VeRO on **TerminalBench-2**, an agentic coding benchmark of 89 long-horizon terminal tasks. VeRO's optimizer improved the baseline agent from 30.3% to 33.7% (+3.4%) through structural code fixes identified via VeRO's observability tools. A second run using raw filesystem access instead of VeRO's structured tools — with the same optimization budget — achieved zero improvement, validating the value of VeRO's design. Full details are in our response to Reviewer QdqQ.
> > >
> > > We also note the planned revisions from our initial rebuttal: GEPA comparison, budget ablation, split rationale table, GPTSwarm/DyLan citations, token/cost reporting, and clarified optimizer-vs-target framing. We are committed to a thorough camera-ready revision.

---

### Official Review · Reviewer_QdqQ · 2026-03-10

**Soundness:** 2
**Presentation:** 3
**Significance:** 2
**Originality:** 4
**Overall Recommendation:** 4
**Confidence:** 4

**Summary:**

LLM agents are becoming the main mechanism for realizing LLM's impact in production. While Reinforcement Learning, the broad gradient-based approach, is well-known, automatic optimization of the agent program itself is underexplored. This paper proposes VeRO (Versioning, Rewards, and Observations) harness that aims to standardize the agent program optimization task as a code generation task. This paper also presents the Agent Optimization Benchmark that contains a standardized suite of target agents, tasks, and evaluation procedures covering multiple domains as a baseline for such an optimization task. Under VeRO, the authors provide several findings in the optimization task.

**Compliance With Llm Reviewing Policy:**

Affirmed.

**Final Justification:**

I raise my score to 4 after seeing Vero's improvement over the baseline on Terminal Bench 2. Automatic agent optimization is a timely problem, and the rebuttal made me more positive about the work.

**Key Questions For Authors:**

1. What does "isolation environments, resource constraints, and guardrails" mean for VeRO?
2. Does the "Git Worktree" commit every time_ there is a file change? That doesn't seem like the right approach to Git, as human engineers might only make a commit when there is a _semantic_ change on the program itself.
3. In section 4.1, why choose an evaluation budget of 8? Have you tried other budgets? Does the target agent's performance plateau at around evaluation budget=8?
	- How does the target agent's performance change during the optimization? Have the authors observed certain trends?
		- The paper mentions the evaluation budget to be 8 evaluation function queries. Do the authors require the agent to stop optimizing when it uses up the budget? Or can the optimizer agent submit early? Have the authors observed the optimizer agent actually degrade the performance of the target agent before it used up the budget?
4. What are the exact code changes in the target agent's implementation while it's being optimized?
5. In section 4.1, why do the authors choose not to evaluate on the code generation task? For example, SWE-Bench's PR and unit tests are great feedback sources for the optimizer to optimize the target agent.
6. In Table 2, it does not seem like VeRO produces much promising improvement with its optimization framework on benchmarks like GPQA and MATH, compared with the baseline. Is there a reason for such results?
	1. Another seemingly missing baseline is how well an agent optimized by methods mentioned in related works performs on the same tasks, to show the necessity of VeRO. For example, the authors can include a *Default-GEPA-optimized* baseline for a baseline optimized by the GEPA algorithm[1].
7. In Table 3, it seems like some results are missing? For example, the pair (Codex, Gemini-Flash) is evaluated on GAIA but not GPQA. Is there a reason why that pair is not included in GPQA?
8. In section 4.3, how do the authors even measure prompt length to be "25-line"? What does "25-lines" of prompt even mean? How many words are there in 25 lines? Same questions for the later "140-line" prompt.
	1. Later in the paragraph, what does "extended libraries (pandas, numpy)" mean?
	2. What does "complex file support" mean? How/Why is it more "complex" than the standard file support?
9. In section 5.4, the authors mention that prompt modifications dominate. Is there an ablation where the optimizer agent can only change the prompt?
[1] GEPA: Reflective Prompt Evolution Can Outperform Reinforcement Learning. https://arxiv.org/abs/2507.19457

**Limitations:**

Yes

**Strengths And Weaknesses:**

# Strengths
- Using LLMs to optimize agents is a timely problem
- Optimizing agents as one coherent software is a novel idea
- The VeRO framework provides a foundation to build agent optimization problem upon.
- VeRO acts as a good example for the community to think about optimizing both the agent's prompts and the agent program itself.
# Weaknesses
- Code generation as a task is not evaluated by VeRO
- The results show that even enabling the optimizer agent to change both the prompts and tools of the target agent, most changes are still on the prompts, questioning the necessity of VeRO over other prompts optimizing baselines.
- VeRO's improvement over the baseline on some tasks is minimal.

---

> ### Author Rebuttal · Authors · 2026-03-30
>
> We thank the reviewer for recognizing the novelty of optimizing agents as coherent software and VeRO as a foundation for the community to build on.
>
> **Q1:** These correspond to 3 of the 6 requirements in Section 3.2: (1) **Isolation environments**: each run operates in a subprocess in a fresh Git worktree with dependency locking via `uv` lockfiles, preventing contamination from untracked artifacts. (2) **Resource constraints**: budget enforcement via the gated *ExperimentRunner* that tracks and blocks evaluation calls exceeding B. (3) **Guardrails**: the *Filesystem* abstraction enforces pattern-based access control, preventing modification of evaluation code and critically preventing the optimizer from reading test split data.
>
> **Q2:** Yes, by design. The auto-commit hook fires on every file write, producing a fine-grained, immutable trajectory. This is not intended to mirror human Git workflows — it serves a distinct purpose: enabling versioned evaluation, rollback to any intermediate state, computing diffs between arbitrary points, and post-hoc trajectory analysis. Algorithm 1 selects among commits that were evaluated, but the full auto-commit history is preserved — this is what enables the interpretability analyses in Section 5.4 (change-type tagging, entropy analysis, diff embeddings). This pattern has precedent — e.g., Jujutsu (jj) automatically records every working-copy change as a revision, treating the full edit history as a first-class object.
>
> **Q3 (New Experiment):** We ran a budget ablation (B ∈ {2, 4, 8, 16, 32}, 3 iterations each):
>
> | Budget | GAIA Val Mean | GPQA Test Mean | MATH Test Mean |
> |---|---|---|---|
> | 2 | 0.119 | 0.627 | 0.890 |
> | 4 | 0.184 | 0.617 | 0.892 |
> | 8 | 0.195 | 0.610 | 0.897 |
> | 16 | 0.184 | 0.613 | 0.896 |
> | 32 | 0.170 | 0.630 | 0.890 |
>
> GAIA peaks at B=8–16 and *declines* at B=32. MATH and GPQA are flat regardless of budget, confirming near-zero gains reflect a fundamental ceiling, not insufficient budget. B=8 means 8 full-dataset evaluations (50–100 instances each), totaling 400–800 agent invocations per run. Figure 7 corroborates: for tool-use tasks, the best commit is found before the halfway point.
>
> **Q4:** Optimization trends are captured in Section 5.4. (1) **Early phases produce the largest gains** — Figure 7 shows the optimal commit is found before the halfway point for tool-use tasks. (2) **Change diversity decreases over phases** — Figure 9 shows entropy dropping sharply after Phase 1, as optimizers revert to prompt-centric changes when structural modifications fail. (3) **Solutions converge semantically** — Figure 10 (UMAP of diff embeddings) shows final commits clustering tightly. **Degradation** is observable in trajectory figures (Appendix Figures 11–14), where scores rise, peak, and decline within a single run. The optimizer can stop early but is prompted to use the full budget. VeRO's safeguard is returning the best-validation commit rather than the final commit (Algorithm 1). Concrete code changes are categorized in Section 5.4/Figure 4; examples from Appendix A.3.2 include adding a Wikipedia search tool (+11.5% GAIA) and adding a verification tool (+5.75% GAIA but -17.8% SimpleQA).
>
> **Q5:** In VeRO, the *optimization task is itself a coding task* — the optimizer reads Python, diagnoses failures from traces, writes tools, and edits prompts. The *target tasks* need not be coding tasks. SWE-Bench was excluded for practical reasons: single-instance evaluation takes 5–15 minutes, so at B=8 a single optimization run would take 40–120 hours. VeRO is task-agnostic and can accommodate SWE-Bench with larger compute budgets.
>
> **Q6:** Near-zero improvement on GPQA/MATH is a *finding*, not a limitation. MATH baseline is 0.87 (ceiling effect); reasoning tasks are bottlenecked by LLM capabilities, not scaffolding. Our budget ablation (Q3) confirms this is not a budget issue. To address the missing baseline, we ran GEPA (reflective prompt evolution, Sonnet 4.5, 3 runs/task):
>
> | | GAIA | GPQA | MATH | SimpleQA | Tau-Bench | Avg |
> |---|---|---|---|---|---|---|
> | GEPA | .21 (.23) | .58 (.61) | .86 (.89) | .66 (.70) | .38 (.42) | .54 (.57) |
>
> GEPA (0.54 avg) performs nearly identically to *Resources Only* (0.54, Table 2) — both restricted to resource-level changes — and well below VeRO Default (0.61), confirming broader code changes provide meaningful gains.
>
> **Q7:** Computational constraints required prioritization. We focused complete coverage on GAIA and extended to other tasks for the most informative optimizer×model pairs. **Revision:** We will clarify the rationale in the table caption.
>
> **Q8:** Lines = newline-delimited lines (Table 10). "Extended libraries" = pandas, numpy, statistics beyond Pawn's standard library. "Complex file support" = Knight parses PDF, Excel, Word, ZIP; Pawn handles plain text only. **Revision:** We will clarify in the camera-ready.
>
> **Q9:** See Q6 — *Resources Only* is this ablation.

---

> > ### Author Rebuttal · Reviewer_QdqQ · 2026-03-31
> >
> > Thank you authors for your rebuttals. Most of my concerns and questions are resolved besides one.
> >
> > Regarding not evaluating on SWE-Bench. I acknowledge that computational constraint. I would really like to see how VeRO perform on coding task. Coding task is an area that LLM is making the most impact on our world right now, and if VeRO can demonstrate performance gains on coding task, I will be even more convinced. If the authors can run some sort of experiments on coding task, that would be great. I am open to any coding benchmarks beside SWE-Bench, such as its derivatives, as long as the authors can handle the computational constraint.

---

> > > ### Author Response · Authors · 2026-04-07
> > >
> > > We appreciate this specific and actionable suggestion. During the discussion period, we ran VeRO on **TerminalBench-2** [1], a competitive agentic coding benchmark of 89 tasks requiring long-horizon, fully autonomous execution in sandboxed terminal environments — spanning scripting, system administration, data processing, and software engineering. TerminalBench-2 is actively contested, with multiple teams engineering harnesses for it, and was already cited in our related work (Section 2).
> > >
> > > **Setup.** Following the benchmark's standard protocol (same as Meta-Harness, Lee et al. [2] — *"the benchmark is small and expensive enough that introducing a separate split would materially weaken the search signal"*), all 89 tasks are used for both optimization and evaluation — a discovery setting where the goal is to find a harness configuration that improves performance on a hard, publicly contested benchmark. We ran two optimization runs on the Terminus-KIRA [3] agent (Haiku 4.5) using the same optimizer model (Sonnet 4.5, effort=high), varying the information access mode:
> > >
> > > | | Run 1 (VeRO tools) | Run 2 (filesystem artifacts) |
> > > |---|---|---|
> > > | Information access | ExperimentViewer + DatasetViewer MCP tools | Raw filesystem (`\_vero/traces/`, `\_vero/datasets/`) |
> > > | Budget | 89 samples | 89 samples |
> > > | Final evaluation | 89 samples | 89 samples |
> > >
> > > Run 2 serves as a controlled baseline: it holds the optimizer model, target agent, benchmark, and evaluation budget fixed, isolating VeRO's structured tool-based information access as the sole treatment variable.
> > >
> > > **Results.**
> > > | | Baseline | Run 1 (tools) | Run 2 (filesystem artifacts) |
> > > |---|---|---|---|
> > > | **Pass rate** | **30.3%** (27/89) | **33.7%** (30/89) | **30.3%** (27/89) |
> > > | Error rate | 46.1% (41/89) | 36.0% (32/89) | 30.3% (27/89) |
> > > | Lines changed | — | 31 | 58 + scripts |
> > > | Delta | — | **+3.4%** | **+0.0%** |
> > >
> > > **Run 1 (VeRO tools)** produced a +3.4% improvement with 31 lines of minimal, targeted changes. The optimizer used VeRO's `ExperimentViewer` to analyze per-sample error breakdowns from the baseline, identified that 11 tasks (12.4%) crashed immediately due to an `AttributeError` in the tool-call parser (the LLM occasionally returned strings instead of dicts), and added `isinstance` guards — completely eliminating the crash class. It also halved the output buffer (30K \\(\to\\) 15K bytes) to address context-window exhaustion and added concise efficiency guidelines. Two of three changes are code-level, not prompt-level.
> > >
> > > **Run 2 (filesystem artifacts)** made more extensive changes — prompt rewrite, loop detection, aggressive context unwinding, analysis scripts — with the same effective optimization budget but 2.6\\(\times\\) the runtime. Its intermediate evaluations on targeted subsets showed 41–45% pass rates, suggesting progress. But the gains vanished on the full test set: 14 previously-passing tasks began failing silently, likely due to the prompt rewrite altering behavior on tasks that were already solved. The net result was zero improvement despite substantially more time and more extensive modifications.
> > >
> > > **Key insight: structured observability leads to better optimization.** Both runs had the same optimization budget (89 samples) and the same optimizer model — the only difference was information access. Run 1's `ExperimentViewer` surfaced per-sample error breakdowns and failure categories on demand, enabling the optimizer to identify the single highest-impact failure mode and write a targeted fix. Run 2's optimizer, reading raw trace files from the filesystem, consumed 1.9\\(\times\\) more input tokens but spread its attention across many failure modes simultaneously, leading to over-engineering and regressions on previously-passing tasks. This directly validates the paper's design choice: structured execution traces and budget-controlled evaluation yield more effective optimization than unstructured access.
> > >
> > > **Why this matters for the paper's thesis.** The most impactful change was a *structural code fix* (crash bug elimination), not a prompt edit — underscoring the paper's central finding that optimizing agents as coherent software yields gains that prompt-only methods cannot capture. The Run 1 vs. Run 2 contrast further demonstrates that VeRO's observability framework is not merely infrastructure but an active contributor to optimization quality.
> > >
> > > **Revision:** We will add the TerminalBench-2 experiment to the paper with full results, error breakdowns, and code diffs.
> > >
> > > [1] Terminal-Bench: Benchmarking Agents on Hard, Realistic Tasks in Command Line Interfaces, Mike A. Merrill et al.
> > >
> > > [2] Meta-Harness: End-to-End Optimization of Model Harnesses, Yoonho Lee and Roshen Nair and Qizheng Zhang and Kangwook Lee and Omar Khattab and Chelsea Finn
> > >
> > > [3] Terminus-KIRA, https://github.com/krafton-ai/KIRA

---

### Official Review · Reviewer_A9Sp · 2026-03-12

**Soundness:** 3
**Presentation:** 3
**Significance:** 4
**Originality:** 2
**Overall Recommendation:** 5
**Confidence:** 5

**Summary:**

VeRO introduces an evaluation framework for agent optimization: the task of using a coding agent to iteratively improve another target agent through edit-execute-evaluate cycles. The paper contributes (1) a framework providing versioned snapshots, budget-controlled evaluation, and structured execution traces, and (2) a benchmark suite with empirical findings across five tasks, providing a foundation for recent emerging research in agent optimization.

**Compliance With Llm Reviewing Policy:**

Affirmed.

**Final Justification:**

I think the paper makes a timely contribution to a newly forming research area - agent harness optimization. The rebuttal addresses most of my concerns.

**Key Questions For Authors:**

1. Can you provide evidence for the claim that uncontrolled coding agents "frequently fail to complete optimization runs or report unreproducible results"?
1. How were train/val/test splits determined? For tasks with no validation set, how is the best commit selected?
1. To strengthen statistical significance without re-running full optimization, have you considered repeating only the evaluation step (multiple eval calls per commit) to better estimate each commit's true performance?

**Limitations:**

Limitations are well-discussed in the paper.

**Strengths And Weaknesses:**

### Strengths

**S1.** The problem is well-motivated and timely. Formalizing agent optimization as a benchmarkable task is a genuine contribution, and the formalism in Section 3.1 is clean and useful.

**S2.** The robustness study and case study are well-designed and produce insightful findings — particularly the result that sophisticated agents benefit more from minimal guidance, and that optimization gains do not reliably transfer across model families.

**S3.** The limitations section is well-discussed. The authors clearly identify gaps in budget definition, missing human baselines, and API variability, which strengthens the paper's credibility.

---

### Weaknesses

**W1. Core motivation claim is unsupported.**

The paper's central motivation — "uncontrolled coding agents frequently fail to complete optimization runs or report unreproducible results" — is stated in the introduction but never empirically demonstrated. There is no ablation or experiment showing what happens without VERO's infrastructure. This is the primary justification for the harness, and it rests entirely on assertion.

**W2. Only author-designed baselines are evaluated.**

As a benchmark paper, a core contribution should be enabling comparison of different agent optimization algorithms under unified conditions. However, all optimizer configurations tested are variants designed by the authors. No existing methods (ADAS, AFlow, DSPy, etc.) are evaluated within VERO. This significantly limits the benchmark's representativeness and utility to the community.

**W3. Benchmark tasks are insufficiently agentic.**

Several tasks — GPQA, SimpleQA, MATH — are fundamentally QA or reasoning tasks. The headroom for agent optimization on these is questionable not because optimization methods are weak, but because the tasks don't require complex agentic behavior to begin with. This is reflected in the results: near-zero improvement on MATH and GPQA across all configurations. More genuinely agentic benchmarks should be included, such as SWE-Bench or TerminalBench — notably, TerminalBench is cited in the related work but not included, with no explanation.

**W4. Train/val/test split design is underspecified.**

The splits vary substantially across tasks (e.g., GPQA has 98 train / 0 val / 100 test, SimpleQA has 46/45/80), and the rationale for these choices is not discussed. This is a fundamental and difficult problem for the field: different agent optimization algorithms may require very different training set sizes to function, and many existing benchmarks treat all instances as test data, making re-splitting inherently difficult for fair comparison. More discussion of this challenge would be valuable, though we recognize this is a fundamental difficulty for the field rather than a limitation specific to this work.

**W5. Evaluation scale is too small.**

B=8 evaluation calls per run leaves very limited optimization space. With only 8 chances to evaluate a modified agent, the optimizer has little room to explore meaningfully, recover from failed modifications, or pursue multi-step improvements. It is unclear whether the near-zero gains on MATH and GPQA reflect fundamental limits of agent optimization on reasoning tasks, or simply insufficient budget to find improvements. A more comprehensive study with larger budgets is needed to draw reliable conclusions.

**W6. Budget definition is incomplete.**

B is defined solely as the number of evaluation calls, ignoring token consumption, API cost, wall-clock time, and latency. The authors acknowledge this in the limitations, which is appreciated. We note that cost measurement in this setting is genuinely complex — a principled budget metric would need to balance multiple dimensions simultaneously. A multi-objective budget metric, or at minimum a reporting of these secondary costs alongside results, would strengthen the benchmark.

---

> ### Author Rebuttal · Authors · 2026-03-30
>
> We appreciate the reviewer's thorough analysis and recognition of the formalism, robustness study, and limitations sections.
>
> **W1/Q1:** We clarify that the statement is *motivation* for VeRO's design, not a standalone empirical finding. Table 2 provides indirect evidence: on the same optimizer model (Sonnet), VeRO Default (0.61 avg) yields +8% over Claude Code Pure (0.53), reflecting the cumulative benefit of versioning, budget enforcement, and structured feedback.
>
> **New Experiment.** We ran an unrestricted optimization: Claude Code (Sonnet 4.5) optimized the GPQA agent *without* VeRO's infrastructure — no filesystem restrictions, no budget enforcement, no split access control. A judge model (GPT-5.4) evaluated all 258 agent messages against the stated rules and found two categories of violation:
>
> - **Test data leakage:** The optimizer printed individual test samples with ground-truth answers (e.g., `ds['test'][0]`), despite explicit instructions not to inspect test data.
> - **Budget overrun:** The optimizer exceeded the 8-eval train budget, then rationalized the overage in its own summary ("within budget if we count v1 as shared").
>
> These are precisely the failure modes VeRO prevents at the infrastructure level. Prompt-level instructions alone are insufficient — the optimizer violates rules it was explicitly given. **Revision:** We will add example traces in the appendix and revise the introduction to: "Agent optimization without controlled infrastructure risks evaluation contamination, irreproducible states, and lost intermediate progress — failure modes that motivated VeRO's core abstractions."
>
> **W2:** VeRO is explicitly designed to be *scaffold-agnostic*. We ran GEPA [1] as an external baseline on all five tasks (Sonnet 4.5, 3 runs/task). Full per-task results are in our response to QdqQ Q6. GEPA achieves 0.536 avg, nearly identical to our *Resources Only* variant (0.54) and well below VeRO Default (0.61 avg, +7.4%). Both GEPA and *Resources Only* are restricted to modifying designated resources (primarily prompts and tool descriptions), confirming that broader code-level changes provide meaningful additional gains.  Additionally, Claude Code (an external system) is already evaluated in two configurations. The contribution is the (meta)harness and benchmark — any coding agent that preserves traceability can plug in. **Revision:** We will add the GEPA comparison to the paper and include an integration guide for community methods.
>
> **W3:** See our response to EH22 Q4 for the full argument. In short: the optimization task *is* a coding task (the optimizer writes Python, tools, prompts via Git). SWE-Bench/TerminalBench were excluded due to evaluation overhead, not conceptual reasons. Reasoning tasks are included by design — the near-zero improvement on GPQA/MATH is a finding, not a deficiency.
>
> **W4/Q2:** We select the best commit per run based on validation performance (Algorithm 1). Test is evaluated only on baseline and best-validation commits. Splits per task: GAIA 50/87/— (text-only filter of official train, test is private); GPQA 98/—/100 (from 198 GPQA Diamond samples, no val split — best commit selected on train performance); MATH 59/60/486 (AFlow splits, re-engineered for reproducibility); TAU-Bench 100/20/115 (standard splits); SimpleQA 46/45/80 (filtered from ~800 to 171 samples where GPT-4.1, Sonnet 4.5, Gemini 2.5 Pro all failed, then randomly split). GAIA and GPQA were not split three ways because the filtered sizes (137, 198) were insufficient — e.g., splitting GAIA three ways (40/47/50) would leave evaluation sets where single-sample differences dominate scores; splitting GPQA's 98 train further (50/48) would give too few science questions across domains. All splits are deterministically fixed and will be released.
>
> **Q3:** Two sources of variance: (1) optimizer stochasticity (captured by N=3 repeated iterations) and (2) evaluation stochasticity (same commit scoring differently on re-evaluation). Each evaluation already averages over 50–100 instances at temperature 0, reducing per-instance variance. Repeating evaluation of the best commit k times would decompose these components. **Revision:** We will adopt this as a recommended protocol extension.
>
> **W5:** See QdqQ Q3 for our budget ablation (B ∈ {2,4,8,16,32}). MATH and GPQA are flat regardless of budget, confirming near-zero gains reflect a fundamental ceiling, not insufficient budget.  We also note an important scale clarification: \\(B=8\\) means 8 full-dataset evaluations (50–100 instances each), totaling 400–800 target agent invocations per run.
>
> **W6:** Acknowledged in Appendix A.4. A principled multi-objective budget metric is non-trivial — as the reviewer notes, cost measurement in this setting is genuinely complex.
>
> [1] GEPA: Reflective Prompt Evolution Can Outperform Reinforcement Learning, https://arxiv.org/abs/2507.19457

---

> > ### Author Rebuttal · Reviewer_A9Sp · 2026-04-06
> >
> > I thank the authors for their detailed response. I believe most of my concerns are addressed. I will increase my score accordingly.

---

> > > ### Author Response · Authors · 2026-04-07
> > >
> > > We thank Reviewer A9Sp for the thorough engagement and for confirming that our responses adequately addressed their concerns. We will incorporate all committed revisions in the camera-ready version.

---

### Decision · Program_Chairs · 2026-04-30

**Decision:**

Accept (regular)

**Comment:**

The paper introduces an evaluation harness and benchmark for agent optimization to systematically study coding agents. An agent is viewed as a piece of code (albeit with different considerations than traditional software engineering), and that code can be optimized via coding agents. The problem is well-motivated and timely, the proposed benchmark is sound (budget-controlled evaluation, versioned agent snapshots, structured execution traces), the exposition is clear. The reviewers flagged issues around budget definition and the limited run counts. The authors sufficiently addressed these issues in their rebuttal with additional ablations (TerminalBench-2 coding experiment, GEPA baseline, budget ablation) that validated their design choices, and the reviewers acknowledged that their main concerns were addressed.